# Immunotherapy with Checkpoint Inhibitors for Hepatocellular Carcinoma: Where Are We Now?

**DOI:** 10.3390/vaccines8040578

**Published:** 2020-10-02

**Authors:** Francesco Tovoli, Stefania De Lorenzo, Franco Trevisani

**Affiliations:** 1Division of Internal Medicine, Azienda Ospedaliero-Universitaria di Bologna, via Albertoni 15, 40138 Bologna, Italy; francesco.tovoli2@unibo.it; 2Department of Medical and Surgical Sciences, University of Bologna, 40138 Bologna, Italy; 3Oncologia Medica, Azienda Ospedaliero-Universitaria di Bologna, via Albertoni 15, 40138 Bologna, Italy; stefania.delorenzo@libero.it; 4Semeiotica Medica, Azienda Ospedaliero-Universitaria di Bologna, via Albertoni 15, 40138 Bologna, Italy

**Keywords:** hepatocellular carcinoma, immunotherapy, checkpoint inhibitors, nivolumab, atezolizumab, bevacizumab, cabozantinib, liver cirrhosis, outcome

## Abstract

Immune checkpoint inhibitors (ICIs) are beginning to show promise in the clinical management of hepatocellular carcinoma (HCC). Most recently, the anti-programmed death protein-1 (PD-1) agent atezolizumab combined with bevacizumab demonstrated superiority to sorafenib in a Phase 3 randomised clinical trial in the frontline setting. Other ongoing trials of immunotherapy for HCC are exploring different drug combinations, such as a double checkpoint blockade with PD-1 and anti-Cytotoxic T-lymphocyte-associated protein 4 (CTLA-4) agents or with tyrosine kinase inhibitors. Moreover, ICIs are being tested in the adjuvant and neoadjuvant settings trying to resolve long-time unmet needs in HCC. The results of the ongoing trials will be critical to understanding the extent of the therapeutic role of ICIs in the complex and multifaceted clinical scenario of HCC. Still, there are some critical points which need further attention to clarify the best use of ICIs in HCC patients. For instance, the actual eligibility rate of patients in the real-life scenario, the prompt identification and correct management of immune-mediated adverse events, the identification of biomarkers predicting response or resistance, and strategies to prevent the tumour escape from ICI effect.

## 1. Introduction

Hepatocellular carcinoma (HCC) is one of the leading causes of cancer-related morbidity worldwide, and the majority of HCC cases occur in a background of chronic liver inflammation [1]. Patients with advanced HCC had no effective therapies until 2008, when sorafenib, a multitarget tyrosine kinase inhibitor (TKI), demonstrated a benefit compared with placebo in terms of both overall survival (OS) and time to progression (TTP) [2]. Even if different frontline and second-line treatments with similar mechanisms of action (lenvatinib, regorafenib, cabozantinib, ramucirumab) have been identified as effective since 2016 [3], the research of drugs inhibiting other tumour pathways remains a top priority [4].

This priority is determined by multiple unmet needs. First, the disease control rate obtained with TKIs rarely exceeds 50–60%, with an objective response rate constantly <10%. Indeed, a relatively high number of patients show a progression at the first imaging evaluation with the practical failure of obtaining tumour downstage or neoadjuvant/adjuvant strategies. Indeed, a high number of patients progress after the first imaging evaluation with the practical failure of obtaining tumour downstage or neoadjuvant/adjuvant strategies. Second, even patients achieving a disease control will experience a relatively short progression-free survival, limiting the possibility of achieving long-term survivals in a sizeable number of patients. Third, most patients will experience TKI-related adverse events (AEs) impairing their quality of life and adherence to the treatment strategy. Moreover, as the safety profile of TKIs widely overlaps between different molecules, patients who are intolerant to sorafenib are theoretically prone to experience similar AEs if treated with other TKIs. As an example, the registrative trial of regorafenib explicitly excluded patients who did not tolerate sorafenib at a reduced dose of 400 mg/day. Consequently, most regulatory agencies approved the use of regorafenib only for patients who progressed under sorafenib, but not for those intolerant to the drug.

The presence of tumour-infiltrating lymphocytes expressing programmed death protein-1 (PD-1) in HCC mass and their correlation with outcome suggested that immunotherapeutic approaches could play a protective role against the progression of this cancer [5,6], as reported for other tumours [7,8]. Immune checkpoints are a normal part of the immune system. Their role is to prevent hyperimmune responses leading to tissue damage. The most known immune checkpoint are PD-1 and Cytotoxic T-lymphocyte-associated protein 4 (CTLA-4). In the oncological setting, pathological activation of PD-1 by its ligands, in particular ligand 1 (PD-L1), can result in the immune escape of the cancer cells. Thus, preventing the activation of the PD-1 receptor can restore the ability of immune cells to recognise and kill tumour cells [9,10]. On the other hand, CTLA-4 is mainly expressed on T cells and regulates the proliferation of activated lymphocytes. In physiological conditions, CTLA-4 regulates the end of the T-cell activity and prevents an excess in T-cell responses. Instead, in pathological conditions, it inhibits the activation, proliferation and production of tumour antigen-activated T cells in the tumour microenvironment (TME) [9,10]. Over time, many different PD-1, PD-L1, and CTLA-4 inhibitor agents have been developed that, collectively, are known as “immune checkpoint inhibitors” (ICIs) (Figure 1).

Despite early promising results in HCC, the first significant Phase 3 trial testing the PD-1 inhibitor nivolumab vs. sorafenib in 2019 failed, as this ICI failed to demonstrate superiority to the TKI [11]. However, only a few months later, a combination of the PD-ligand 1 (PD-L1) atezolizumab with the monoclonal anti-VEGF (vascular endothelial growth factor) antibody bevacizumab significantly increased OS in comparison with sorafenib, ending a 12-year history of failures in searching therapies able to outperform sorafenib [12]. Indeed, previously only another TKI (lenvatinib) had succeeded in substantially challenging sorafenib in a noninferiority trial, but without reaching the threshold of a statistically significant superiority [13].

The positive result of atezolizumab-bevacizumab combination projects a favourable light on multi-agent immunotherapy trials that are ongoing for HCC. At present, ICIs are being tested as PD1/PD-L1 monotherapy, in combination with anti-VEGF agents, paired with TKIs, or with other therapeutic agents. Their window of utility is no more restricted to the advanced setting, but is expanding in the adjuvant and neoadjuvant setting. Nevertheless, their rapid appearance and expansion in the clinical scenario have left some unanswered questions which will represent the hot topics of the future HCC treatment. This review aimed to guide the reader across the multiple existent trials and to present the scenarios will be opened by immunotherapy in the years to come.

## 2. Materials and Methods

This narrative review was written including oversight of concluded clinical trials on ICIs for HCC treatment, which incorporates both articles published in extensor and oral or poster communications presented in international scientific congresses. We checked MEDLINE with different strings, including the name of the various ICIs in combination with the MeSH term “carcinoma, hepatocellular”. Moreover, we searched the online libraries regarding congresses of the following scientific associations: American Society of Clinical Oncology, American Association for the Study of Liver Diseases, European Society of Medical Oncology, and European Association for the Study of the Liver. As a third step, we searched ClinicalTrials.gov (accessed on 31 May 2020) to identify all the ongoing clinical trials concerning HCC. As a result, 486 clinical protocols were analysed, and all trials of ICIs were included in this review. Finally, the part dealing with critical points analyses information retrieved in articles identified in MEDLINE and papers known to the attention of the Authors for their scientific relevance.

## 3. Results

### 3.1. Concluded and Ongoing Trials

#### 3.1.1. Anti-PD1 Inhibitors as Single Agents

Following the promising results of an investigator-initiated Phase 2 open-label trial on the CTLA-4 inhibitor tremelimumab in HCC patients [14], the worldwide attention of hepatologists toward ICIs was increased by the announcement of the results of a multicohort Phase 1b/2 trial exploring the effect of PD1-inhibitor nivolumab (Checkmate-040, NCT01658878). The interest was bolstered as this trial reported an unprecedented overall response rate (ORR) of 15% across the dose-escalation and expansion cohorts [15]. Moreover, a Phase 2 trial assessing the role of pembrolizumab in patients who had progressed or were intolerant to sorafenib (Keynote-224) showed similar encouraging results [16]. Other preliminary results came from Phase 1b/2 trials with camrelizumab and durvalumab, confirming an objective response rate of 15–20% of cases with a disease control rate >50% [17,18].

The expectations deriving from these results were subsequently forestalled by the failure of two large Phase 3 trials comparing nivolumab vs sorafenib and pembrolizumab vs placebo in the frontline and second-line setting, respectively. Indeed, in the CheckMate-459 study, 743 patients naïve to systemic treatments were randomised 1:1 to nivolumab or sorafenib [11]. The median OS was 16.4 months for nivolumab and 14.7 months for sorafenib (hazard ratio [HR] 0.85; 95% confidence interval 0.72–1.02; *p* = 0.0752) [11]. While the survival in the nivolumab arm was the highest ever reached for a frontline systemic drug in clinical trials of HCC, the surprisingly high survival of the sorafenib arm prevented nivolumab from reaching a statistically significant OS bonus. Different reasons have been advocated to explain this unexpected performance of sorafenib in the CheckMate-459 study, considering that survival in the treatment arm of the registrative SHARP trial was 10.7 months [2]. They include overtime improved tailored management of patients undergoing sorafenib [19], evolving stage-migration strategy with a higher number of patients being treated in the intermediate-stage of HCC, and a high proportion of patients receiving a second-line systemic treatment after sorafenib discontinuation [11]. Despite its failure, the CheckMate-459 trial provided interesting data about the safety of ICI therapy in a large population, indicating that the safety profile of nivolumab was more favourable than that of sorafenib, with a rate of G3-G4 adverse events (AEs) of 22% vs. 49%, and a rate of discontinuation, due to AEs, of 4% vs. 8% for nivolumab and sorafenib, respectively [11]. Moreover, the quality of life was better in the nivolumab treatment arm. Lastly, this trial confirmed an ORR of 15% for nivolumab.

In the KEYNOTE-240 trial, 413 patients who failed sorafenib, were randomised 2:1 to pembrolizumab or placebo [20]. Median OS was 13.9 months (95% confidence interval, 11.6–16.0) for pembrolizumab vs. 10.6 months (95% confidence interval, 8.3–13.5) for placebo (HR 0.781; 95% confidence interval, 0.611–0.998; *p* = 0.0238). In parallel, the median progression-free survival (PFS) was 3.0 months vs 2.8 months (HR, 0.718; 95% CI, 0.570 to 0.904; *p* = 0.0022) for pembrolizumab and placebo, respectively. Although OS and PFS improved compared with placebo, their respective HR did not meet the pre-specified boundaries of *p* = 0.0174 for OS and *p* = 0.0020 for PFS. Similar, in the CheckMate-459 study, the survival of the treatment arm was the highest ever reached in a second-line setting—but these good results were statistically nullified by an unexpectedly high survival in the control arm. In this study, the relatively low number of patients enrolled, and the high number of placebo patients (47.4%) receiving post-study chemotherapies were blamed for the failure of the trial. In agreement with the CheckMate-459 study, the ORR for pembrolizumab was 18.3%, and the safety profile was good, with similar rates of Grade ≥3 AEs across groups (52.7 vs. 46.3% in the pembrolizumab and placebo arms, respectively). Instead, the rates of AST, ALT, and bilirubin increase were higher in the treatment arm (22.6% vs. 13.3%, 17.6% vs. 6.1%, and 18.6% vs. 7.5%, respectively). As a result, discontinuation of treatment because of AEs occurred in 48 patients (17.2%) in the pembrolizumab arm and 12 (9.0%) in the placebo arm. Interestingly, the AEs and discontinuation rates were more frequent with pembrolizumab than with nivolumab, probably underlining the frailer characteristics of patients enrolled in the second-line setting. Pertinently, in the Checkmate 459 trial, a high proportion of placebo patients discontinued the treatment due to the occurrence of AEs, which of course, were caused by the underlying clinical condition.

Overall, the results of the CheckMate-459 and KEYNOTE-240 trials suggested that: (1) The design of clinical trials remains a critical point in HCC trials, as seemingly minor pitfalls can determine the failure of the trial [21]; (2) the safety profile of nivolumab and pembrolizumab was excellent, with an impact on the quality of life significantly lower than that of sorafenib in the CheckMate-459 study [11]; (3) an objective response could be achieved in 15–20% of cases, leading to a long-term survival in responding patients [11,20].

Consequently, the attention toward ICIs for HCC treatment remained high and suggested switching from monotherapies to combination strategies. As a consequence, only one Phase 3 trial comparing a single agent tislelizumab vs sorafenib is currently ongoing [22]. Its enrolment closed in 2019, and the results are still pending. Although this agent is very similar to nivolumab, its success is theoretically possible, given the proximity of the significance threshold reached by nivolumab, the slightly different trial design, the accumulated experience with ICI therapy in HCC, and the high intrinsic heterogeneity of these patients.

#### 3.1.2. Combination of PD1 Inhibitors with Intravenous Anti-VEGF Agents

Shortly after the delusional results of the CheckMate-459 study [11], a clinical trial demonstrated the superiority of combination therapy versus sorafenib. The IMBrave150 study was a global Phase 3 clinical trial in which 501 patients were randomly assigned to the combination of atezolizumab-bevacizumab versus sorafenib in a 2:1 fashion [12]. The rationale of combining an anti-VEGF2 monoclonal antibody with a PD-L1 inhibitor relies on a possible synergistic effect of the two agents. In particular, the VEGF-blockade may enhance the anti-PD-L1 efficacy by reversing VEGF-mediated immunosuppression and promoting T-cell infiltration in tumours [23,24]. The enrolment was stopped following an interim analysis showing the superiority of the combination treatment. Slightly more than half of the scheduled participants had been randomised at the time of this stop, with a median follow-up duration of 8.6 months. The median OS was not reached in the atezolizumab-bevacizumab arm after 17 months, while it was 13.2 months in the sorafenib arm. The estimated rates of survival at 6 and 12 months were 84.8% and 67.2%, respectively, in the atezolizumab–bevacizumab group, and 72.2% and 54.6% in the sorafenib group. The median PFS was significantly longer in the combination treatment arm than in the sorafenib arm (6.8 months (95% confidence interval 5.7–8.3) vs. 4.3 months (95% confidence interval, 4.0–5.6)). According to independent assessment with RECIST 1.1 criteria, the ORR was 27.3% with atezolizumab–bevacizumab and 11.9% with sorafenib (*p* < 0.001). Overall, the toxicity of the combined therapy was manageable. Still, the safety profile differed from that of the monotherapy with ICIs. The most common AE was arterial hypertension (15%), a known bevacizumab-related AE [12], and the rate of serious AEs was slightly higher in the combination than in the sorafenib arm (38.0% vs. 30.8%), as well as the proportion of patients permanently discontinuing treatment for toxicity (15.5% vs. 10.3%). However, no specific events were identified as responsible for the increased incidence of serious AEs. Interestingly, the rate of aspartate (AST) and alanine (ALT) aminotranspherases and bilirubin increases was similar in the two arms. Finally, the combined therapy resulted in longer clinical deterioration times of patient, and a better quality of life than sorafenib.

Currently, no other Phase 3 trials are evaluating the combination of ICIs and intravenous anti-VEGF agents in the advanced setting of HCC. Most Phase 1 and 2 clinical studies are investigating the association with bevacizumab, while one study is testing tivozanib, another anti-VEGF agent (Table 1). Notably, there are no RCT testing ICIs plus ramucirumab, the only anti-VEGF agent which has demonstrated a specific activity as a single agent in advanced HCC (albeit only in the subgroup of patients with high alfa-fetoprotein).

#### 3.1.3. Combination of PD1 Inhibitors with Tyrosine Kinase Inhibitors

The hypothesis of combining TKIs and ICIs has been considered another possible strategy to treat HCC. This choice gained further strength after the failure of the CheckMate-459 trial [11]. Similar to the combination of ICIs and intravenous anti-VEGF agents, ICIs and TKIs could have synergistic effects, as molecular target agents could collectively block the signalling from various growth factors and affect immune effectors, as well as the tumour vasculature [25,26].

Until now, no registrative trials testing this combination have been concluded, but four ongoing Phase 3 RCTs planned in the first-line setting testify the interest toward the combination ICIs/TKI (Table 2).

Two of these Phase 3 studies, namely, the 3-arm COSMIC-312 trial comparing atezolizumab-cabozantinib vs cabozantinib vs sorafenib and the 2-arm LEAP-002 trial testing pembrolizumab-lenvatinib vs lenvatinib have closed their enrolment. Overall survival and PFS are the primary endpoints of these studies, whose results are eagerly waited to understand the potentialities of these combinations.

Promising results come from a very recently published Phase 1 study assessing the efficacy of pembrolizumab-lenvatinib combination [27]. Amongst 104 patients who were enrolled in this study, the objective response per RECIST 1.1 criteria was found in 36% of them, the median PFS was 8.6 months and a staggering OS of 22 months is reported. While no new safety signals were detected, Grade ≥3 treatment-related AEs occurred in 67% of patients with three treatment-related deaths.

Some exciting hints also came from the preliminary results of the cohort 6 of the CheckMate-040 trial [28]. In this cohort, 71 patients were randomised to the combination of nivolumab-cabozantinib or the triple combination of nivolumab-ipilimumab-cabozantinib [28]. Median OS was not reached in either arm; median PFS was 5.5 months for the doublet arm and 6.8 months for the triplet arm. Investigator-assessed ORR was 17% in the doublet arm and 26% in the tripled arm. Grade 3–4 treatment-related AEs (TRAEs) were more prevalent in the triplet than in the doublet arm (71% vs. 42%), leading to a higher rate of treatment discontinuation (20% vs. 3%). The authors concluded that, although the triplet regimen had a higher rate of AEs than the doublet one, the majority of these AEs were manageable and reversible, without new alarming signals.

#### 3.1.4. Combination of PD-1 and CTLA-4 Inhibitors (Dual Checkpoint Blockade)

The efficacy and safety of the combination of anti-PD-1 (or anti-PD-L1) and anti-CTLA-4 agents have been tested. Adding the CTLA-4 to the PD-1/PD-L1 blockade can enhance the immune response against the tumour with an increased likelihood of achieving an objective response, but at the price of increased toxicity [29].

A Phase 3 trial (HIMALAYA, NCT03298451) is testing the effects of the combination of tremelimumab-durvalumab vs. durvalumab vs. sorafenib in the frontline setting of HCC. The enrolment was closed several months ago, and the required number of events is likely to close to being reached.

A second Phase 3 trial (CheckMate 9DW, NCT04039607) is comparing the combination of nivolumab-ipilimumab vs sorafenib in the same setting. This trial is the natural continuation of the Cohort 4 of the CheckMate-040 study, which provided interesting results on dual checkpoint blockade (albeit in the second-line setting) [30]. In this cohort, 148 sorafenib-experienced patients received the combination of nivolumab + ipilimumab according to three different schedules: Nivolumab 1 mg/kg Q2W and ipilimumab 3 mg/Kg Q3W for the first three months, followed by nivolumab 1 mg/Kg Q2W (Group A); nivolumab 3 mg/kg Q2W and ipilimumab 1 mg/Kg Q3W for the first three months, followed by nivolumab 3 mg/Kg Q2W (Group B); or nivolumab 1 mg/kg Q2W and ipilimumab 3 mg/Kg Q6W (Group C) (29). Efficacy data showed an unprecedented OS of 22.8 months in Group A, 12.5 months in Group B, and 12.8 months in Group C [30]. The ORR was impressively high in all groups (32%, 31% and 31%, respectively) [30]. It is worth noting that Group A also had the higher rate of AEs, and the proportion of patients permanently discontinuing the treatment for toxicity was 18%, 6% and 2% in the three groups (29). The frequency of AST and ALT increases reached the maximum in Group A (20% and 16%, respectively), similarly to that of “hepatitis” (20%). Moreover, in arms A, B and C, 7 out of 10, 3/6 and 2/3 patients showing immune-mediated hepatic AEs required high-dose glucocorticoids (≥40 mg of prednisone daily or equivalent) for a median period of two weeks [30]. However, no further immune-modulating treatment was required, and 90% of these events resolved following the pre-specified protocol instructions [30].

The choice to switch from the second-line to the frontline setting the combination of nivolumab + ipilimumab after the promising preliminary results reflect a typical orientation of the pharmaceutical industry for HCC treatment with ICIs. Indeed, only one Phase 2 trial is currently investigating, as second-line treatment, a dual checkpoint blockade with sintilimab + ipilimumab (NCT04401813).

#### 3.1.5. Other Therapeutic Combination of Immune Checkpoint Inhibitors

Combination strategies of immune checkpoint inhibitors (ICIs) are not limited to the anti-VEGF drugs and TKIs (Table 3).

In particular, many trials are evaluating combinations with c-MET oral inhibitors (APL-101, capmatinib), anti-phosphatidylserine antibodies (bavituxumab), transforming growth factor-beta oral or intravenous inhibitors (galunisertib and ascrinvacumab), heat shock protein 90 inhibitors (XL-888) or conventional chemotherapy (FOLFOX4).

Of interest, some trials are exploring the mechanisms that may lead to the resistance to ICIs, in an attempt to identify other modulators of the immune response, which can help in overcoming this phenomenon. For instance, EPACADOSTAT and BMS-986205 are two drugs targeting indoleamine 2,3-dioxygenase 1 (IDO-1), which are being tested in combination with nivolumab.

A multi-arm trial of nivolumab as a single agent or in combination with the anti-interleukin-8 agent BMS-986253 or with the anti-colony stimulating factor-1 receptor (CSF1R) drug cabiralizumab is also in progress.

Also, INCAGN01949 is an investigational drug targeting the T-cell costimulatory molecule CD134 which is tested in combination with nivolumab and ipilimumab. Lastly, the combination of nivolumab and ABX196 is supposed to activate the natural killer (NK) T lymphocytes, which can theoretically prevent tumour progression under ICIs due to the loss of human leukocytes antigen (HLA) present on the surface of HCC cells.

#### 3.1.6. Adjuvant and Neoadjuvant Setting

To prevent a recurrence after eradication of HCC with curative resection or locoregional therapy is still an unmet need. Indeed, no drug has so far demonstrated efficacy in this task. In the adjuvant setting, the STORM trial sorafenib failed in showing a superiority on placebo in terms of RFS [31]. Moreover, the low ORR of TKIs makes unsuitable the use of these drugs in the neoadjuvant setting.

ICIs can strengthen the immune attack against residual tumour cells after surgery or locoregional treatments, and their relatively high ORR, mainly when prescribed in combinations, could also help bring tumour burden within the limits of surgical respectability in some patients with intermediate or even advanced HCC [32]. The results of a Phase 1 trial with tremelimumab in combination with ablative procedures provided the critical information on the neoadjuvant use of ICIs [33]. In this study, 32 patients received tremelimumab at two different dosages, followed by subtotal percutaneous ablation on Day 36. Nineteen patients were evaluable for radiological response, and five of them achieved a complete response. No dose-limiting toxicities were found. More intriguingly, tumour biopsies performed six weeks after tremelimumab showed an accumulation of intratumoral CD8+ T cells, indicating a re-activated immune response that could clear residual cancer cells after a subtotal ablation.

Unsurprisingly, the adjuvant and neoadjuvant settings are very competitive scenarios attracting various industries, and several relevant RCTs are currently in progress (Table 4).

Currently, four different Phase 3 trials are evaluating anti-PD1 agents as single agents or in combination with bevacizumab as adjuvant therapies following curative procedures in patients with HCC features indicating a high recurrence risk, such as large nodules, multinodular disease, microvascular invasion, poorly differentiated tumours. Moreover, a Phase 2–3 RCT testing toripalimab and more than 10 Phase 1 or 2 non-randomised studies are undergoing in the adjuvant or neoadjuvant setting of HCC.

#### 3.1.7. Combination with Local Treatments

The combination of locoregional treatments and ICIs is an intriguing option, for at least two reasons: First, a simple additive effect could justify the use of this association in difficult-to-treat cases; second, and more intriguingly, ICIs and locoregional treatments could have a synergistic effect. The liberation of tumour-associated antigens after the tumour destruction by locoregional treatments can lead to the priming of immune cells, a phenomenon which can be theoretically enhanced by ICIs [34]. Multiple studies are, therefore exploring this therapeutic combination (Table 5).

Three Phase 3 RCTs are exploring the efficacy and safety of ICIs in combination with TACE. Sintilimab is tested as a single agent in one study, durvalumab is prescribed in combination with bevacizumab in another one, while the third study will evaluate the doublet pembrolizumab-lenvatinib.

Besides, some Phase 2 and 1 studies are evaluating the effect of ICIs (sintilimab and pembrolizumab) in combination with external beam or selective intra-arterial radiation therapy. Since radiation therapy has been shown to liberate tumour-associated antigens provoking the so-called “abscopal effect” (i.e., a response also in non-treated lesions) in some patients [35], ICIs could theoretically increase this phenomenon leading to responses of unexpected magnitude.

### 3.2. Open Problems

#### 3.2.1. Eligibility in the Real-World Clinical Practice

The target population of the concluded ICI studies on HCC includes patients with an advanced-stage (BCLC-C) cancer or an intermediate-stage (BCLC-B) neoplasm not amenable to surgery or locoregional procedures. However, in the real-world clinical practice, not all of these patients are eligible to receive ICIs as monotherapy or combination, as they may have specific contraindications.

Recently, Giannini et al. [36] explored the Italian Liver Cancer (ITA.LI.CA) database to assess the theoretical applicability of ICIs in field-practice conditions according to the criteria utilised for patient enrolment in clinical trials. The ITA.LI.CA database includes patients with newly diagnosed or recurrent HCC managed in a large number of Italian centres with different levels of specific expertise. This database, due to its heterogeneity in terms of tumour stage, the severity of underlying liver disease and therapeutic approaches, predicts the analysis of the potential utilisation of these drugs. Amongst the 2483 patients (distributed across different BCLC stages), 525 (21.1%) and 268 (10.8%) were theoretically eligible for nivolumab and pembrolizumab, respectively, as frontline therapy [36]. Considering only the 1514 patients in the advanced-stage or the intermediate-stage, but unresponsive to locoregional procedures, the rate of eligibility raised to 34.7% for nivolumab and 17.7 for pembrolizumab. Child-Pugh class >A (*n* = 601), uncontrolled ascites (*n* = 380), performance status >1 (*n* = 343), active alcohol intake (*n* = 323), thrombocytopenia (*n* = 299), hepatic encephalopathy (*n* = 155), aminotransferase levels >5× (*n* = 123) and concurrent autoimmune diseases (*n* = 57) were amongst the main limitations to the potential use of ICIs in the front-line setting (34). The eligibility in the second-line setting was even lower, with 5.4% and 8.0% of patients amenable for nivolumab and pembrolizumab, respectively [36].

We also analysed the same database to verify the applicability of other therapeutic options based on the conventional inclusion/exclusion criteria adopted for the clinical trials of intravenous anti-VEGF agents and TKIs. Overall, 52 additional patients had clinically significant heart disease, ten patients had uncontrolled hypertension, fifteen had chronic non-healing skin ulcerations, and three had non-liver-related coagulative abnormalities increasing the risk of bleeding. Consequently, the rate of patients eligible to the atezolizumab-bevacizumab combination (and, by extension, to a combination of ICIs and TKIs) was 17.9% in the whole HCC population and 29.4% in HCC patients with an advanced HCC or an intermediate tumour not eligible for surgery or locoregional procedures (unpublished data).

Therefore, analyses of a large unselected cohort of HCC patients generated by the real-world clinical practice would indicate that, among potential candidates to immunotherapy, no more than one-third of them are amenable to ICIs as a frontline approach, and this percentage further decreases considering combination therapies with anti-VEGF or TKI agents.

#### 3.2.2. Safety

The inhibition of physiological immune checkpoints may be associated with immune-related AEs (irAEs) targeting the skin, gut, thyroid, adrenal glands, lung and liver [15]. For monotherapies with PD-1 /PD-L1 inhibitors, the risk of irAEs is dose-independent, with an incidence of 27% for all Grades, and 6% for Grade ≥3 [37]. Instead, with CTLA-4 inhibitors, the overall incidence of irAEs is dose-dependent and remarkably higher, reaching 72% for all Grades and 24% for Grade ≥3 [38]. Generally, these events are easily manageable, delaying the subsequent scheduled dose and using corticosteroids in severe or unresponsive cases. A recent meta-analysis reports 42 (0.6%) cases of fatal irAEs among 6528 patients treated with ICIs, with ipilimumab-induced colitis being the leading cause of death [39]. Furthermore, a minimal number of fatal outcomes due to immune-related pneumonitis [40] and myocarditis [41] have been reported.

Despite this acceptable safety profile of ICIs, a justifiable concern on the expected risk/benefit ratio accompanies their use in cirrhotic patients, for different reasons. First, immune-related hepatitis can precipitate an acute-on-chronic liver failure with a high risk of severe liver decompensation and death. Second, the use of corticosteroids to treat severe irAEs is particularly problematic in cirrhosis, especially in terms of increased risk of infections and ascitic decompensation. Third, cirrhosis is known to disrupt the liver’s homeostatic immune function, provoking per se a condition, including both systemic inflammation and immunodeficiency [42]. Indeed, a study enrolling patients treated with ICIs for different cancers seemed to suggest that hepatic AES were related to a worse prognosis [43].

Sangro et al. [14] reported a rate of aminotransferase increase close to 50% in their pivotal trial with tremelimumab. However, these alterations were transient, never associated with liver function impairment, and resolved without corticosteroids [14]. Fortunately, even the safety reports from subsequent clinical trials testing ICIs in HCC patients were reassuring [15,17,18]. Moreover, the large CheckMate459 and KEYNOTE-240 trials confirmed that the safety profile of ICIs was consistent with that reported in previous studies for melanoma and lung cancer [11,20], suggesting that cirrhotic patients have not an increased risk of liver irAEs. The proportion of cases who needed corticosteroid treatment was 6% for durvalumab (18) and 20% for the nivolumab-ipilimumab combination [30]. It is worth noting than the risk of relevant AEs in HCC studies increased when ICIs were tested in combinations with other agents (Table 6).

Overall, the comforting data gave support to the use of nivolumab in Child-Pugh B patients. In these particularly frail subjects, treatment-related hepatic AEs were reported in only 4 out of 49 patients, leading to treatment discontinuation in 2 patients [44].

Nevertheless, both HCC and liver cirrhosis act as confounders for other types of irAEs. Cutaneous toxicities, for instance, are the most common AE reported in clinical trials on ICIs (28). The interpretation of skin toxicities may be difficult when ICIs are prescribed in combination with TKIs as this class of drugs has this potential AE [19] and the hand-foot skin reaction, typical of TKIs, has not been reported for ICIs. It is also pertinent to note that the TKI-related skin rash usually appears during the first week of treatment and quickly disappears after drug discontinuation due to the short half-life of most TKIs [13]. In contrast, ICI-related skin toxicities appear later, and in the absence of steroid therapy, requires a long-lasting interruption of treatment to resolve [39].

Diarrhoea is another common irAE which can difficultly be ascribed to a precise cause. Indeed, cirrhotic patients are often medicated with osmotic laxatives (the dosage of which must be accurately tailored) to prevent hepatic encephalopathy, and in patients treated with both TKIs and ICIs, the same considerations made for dermatological AEs apply to diarrhoea. When diarrhoea is associated with abdominal pain and signs of colonic inflammation, immune-related colitis should be suspected and immediately managed, as it still represents the most frequent cause of death due to ICIs [39]. Although the diagnosis of immune-related colitis is frequently made based on clinical signs and symptoms, colonoscopy is the diagnostic gold standard and assesses its severity and prognosis [45].

Immune-related endocrinopathies also pose challenges. Thyroid function is often monitored in patients with advanced HCC as a result of their inclusion in clinical trials or because of a concurrent treatment with TKIs, which can provoke thyroid dysfunctions [2,13]. As a consequence, immune-related hypo- and hyperthyroidism are usually detected in a pre-clinical phase. On the contrary, the identification of adrenal failure can be problematic as cirrhotic patients have an intrinsic tendency to hypotension due to the hemodynamic peculiarities of advanced liver disease and slight hyponatremia due to haemodilution and use of diuretics.

Other irAEs are unrelated to the underlying liver disease or concurrent therapies, but require immediate attention as a late diagnosis might cause a dismal prognosis. For instance, the appearance of cough, fever and shortness of breath should prompt immediate investigations to detect an immune-related pneumonitis, which requires an early treatment since acute respiratory failure may rapidly ensue [40].

#### 3.2.3. Unpredictable Efficacy, the Need for Biomarkers

All trials of anti-PD-1/PD-L1 for HCC consistently identified a subgroup of 15–20% of patients obtaining an objective response (with an increase of this proportion up to 36% using combination regimes) [27]. These patients also obtained the most important benefit in terms of OS. Therefore, the identification of predictors of response would have a crucial role in optimising the cost-effectiveness of therapy with ICIs. At the same time, predictors of futility might channel patients to other treatments (TKIs, for instance), avoiding the cost and risk of pointless irAES.

Historically, immunostaining of tumour specimen with anti-PD-L1 antibodies was the first approach used to predict the response to ICIs. However, in most studies, PD-L1 expression was not predictive of response. When it was claimed as predictive, different thresholds of PD-L1 were identified in different tumours (from 1% to 50%) [46]. Moreover, the determination of PD-L1 expression suffers from the intrinsic variability of immunohistochemistry [47]. Moreover, the biological characteristics of malignancy, including intratumoral heterogeneity and tumour microenvironment, play an essential role in reducing the reliability of this technique [48]. Notably, Bensch et al. [49] performed the first-in-human study assessing PD-L1 expression by radionuclide imaging (89 Zr-atezolizumab), finding a good correlation between increased tumour uptake and response to anti-PD1 therapy. However, the reliability of this intriguing non-invasive way to detect PD-L1 expression that avoids sample biases needs further validation.

The role of PD-L1 expression has been evaluated in a patient subgroup of the CheckMate-459 study testing nivolumab vs sorafenib. In the nivolumab arm, the OS did not differ between patients with low and high PD-L1 expression. At the same time, surprisingly, the OS was different in the sorafenib arm, being about 14 months in patients with low PD-L1 expression and eight months in those overexpressing PD-L1 [11]. Altogether, these results suggest that PD-L1 expression has a negative prognostic effect in HCC patients, and a PD1 blockade (but not sorafenib) can reverse this negative effect on survival. Nevertheless, the benefit of nivolumab cannot be predicted by PD-L1 expression alone, as it did not affect the OS of patients undergoing this treatment.

Meanwhile, other putative biomarkers are under investigation. Several modern pieces of research are exploring the relationship between DNA damage/mutations and tumour immunogenicity. Tumour Mutational Burden (TMB) is a quantitative measure of the total number of nonsynonymous mutations per coding area of the tumour genome and is considered a surrogate marker of tumour immunogenicity reflecting neoantigen load. TMB is usually calculated using next-generation sequencing (NGS) techniques on tumour samples. Moreover, new blood tests (bTMB), exploring a limited number of genes, are under investigation in an attempt to obtain liquid biopsies in patients with tumours inaccessible to biopsies [50]. TMB determination, however, suffers from the same limitations of PD-L1 staining, namely, the lack of standardised thresholds and variability in quantification methods [51].

Mutations in the mismatch repair (MMR) system and microsatellite instability (MSI) are other DNA alterations potentially associated with increased tumour immunogenicity. In particular, tumours harbouring an erroneous MMR system will accumulate DNA mutations, which can lead to the presence of high levels of mutation-associated neoantigens [52,53], so that anti-PD1 agents are now prescribed to patients with colorectal cancers showing MSI [54]. The role of these DNA alterations in guiding the HCC treatment remains to be established.

#### 3.2.4. Resistance to Immune Checkpoint Inhibitors

Despite the overall encouraging results of immunotherapy, most HCC patients under anti-PD1/PD-L1 and anti-CTLA4 agents eventually experience a disease progression. Resistance to ICIs can be primary or acquired.

The primary resistance to ICIs is due to a paucity (or even lack) of intratumoral immune infiltrate, which suggest a defective immune cell trafficking. Interestingly, this profile of “immune exclusion” is often associated with an activated Wnt/ß-catenin pathway signalling in HCC [55], giving support to the role of Wnt/ß-catenin activation as biomarker predictive of resistance to ICIs.

Primary resistance to ICIs may also derive from a more complex alteration of the immune system, with other immune pathways lying outside of the classical PD-1/PD-L1 and CTLA-4 checkpoints. For instance, lymphocyte activation gene-3 (LAG-3) is involved in the inhibition of CD8+ T cell and NK cell functions. Its expression is associated with a poor prognosis in HCC patients [56]. Moreover, T-cell immunoglobulin and mucin-containing protein-3 (TIM-3) and its ligand galectin-9 can activate a complex cascade that ultimately leads to T-cell exhaustion [56].

It can be argued that LAG-3, TIM-3 and PD-1 act synergistically, facilitating the HCC immune evasion, and could mediate the resistance to the classical PD-1/PD-L1 blockade [57,58]. Some trials are currently investigating the association effects of ICIs combined with TIM-3 (NCT03099109) and LAG-3 (NCT01968109) inhibitors in solid tumours.

Tumour microenvironment (TME) is another possible player in the development of primary resistance to ICIs. TME includes not only immune cells, but also blood vessels, fibroblasts, signalling molecules and the extracellular matrix surrounding the tumour [59]. Indoleamine-pyrrole 2,3-dioxygenase 1 (IDO-1) is a heme-containing enzyme physiologically expressed in many tissues and cells, which is activated during tumour development, helping malignant cells escape eradication by the immune system (56). Tumours with high IDO1 deplete the essential amino acid tryptophan from TME, resulting in T-cell anergy and immune suppression [59,60]. EPACADOSTAT and INCAGN01949 are two drugs targeting IDO and the T-cell costimulatory molecule CD134, which are being tested in combination with ICIs for HCC (NCT02178722, NCT03241173).

The role of TME is strictly related to the phenomenon of the epithelial to mesenchymal transition (EMT), a cellular process that enables epithelial cells to gain mesenchymal features leading to an aggressive and motile phenotype [61]. Several animal models and in vivo patient studies have shown that the activation of EMT in HCC promotes tumour progression and metastasis [62]. Moreover, EMT can promote an immunosuppressive TME by recruitment of tumour-associated macrophages, regulation of immune checkpoint molecules and immune resistance to NK cell-mediated lysis [63,64]. The association between EMT and immunosuppression has been reported in different cancer types, including HCC [65]. The role of tumour growth factor beta (TGF beta) is also of particular interest. This multifunctional cytokine plays multiple key activities, because of its role in immune and stem cell regulation and differentiation [66,67]. In many cancer cells, the TGF-β signalling is disrupted [68], and therefore, TGF-β is no longer able to downregulate the cell cycle, causing a simultaneous proliferation of both cancer and surrounding stromal cells in the setting of an immunosuppressive and pro-angiogenic microenvironment [69]. Additionally, TGF-β can convert effector T-cells into regulatory T-cells [70] and exerts inhibitory effects on B-cells [71], turning off the inflammatory reaction and favouring tumour immune escape. Investigation on the combination of nivolumab and the TGF-beta inhibitors galunisertib and ascrinvacumab (NCT02423343, NCT03893695) is currently in progress and will provide valuable information about the ability of these combinations in overcoming resistance to ICIs.

The VEGF signalling pathway can also provoke immune resistance as it induces Fas ligand, leading to cell death in tumour-infiltrating CD8+ T cells [72]. The results of the ImBrave-150 trial testing atezolizumab-bevacizumab would support the hypothesis of a consistent role of this pathway in HCC progression. However, the design of this trial does not clarify whether the efficacy of atezolizumab-bevacizumab derived from a synergistic or additional effect [73].

Currently, there are no data suggesting that the cirrhotic microenvironment actually affects the efficacy of ICIs. Indeed, when the first trials of immunotherapy for HCC were designed, this hypothesis (deriving from pre-clinical experiences) [42] was considered, but subsequent clinical data showed a similar efficacy both in viral and nonviral patients. Even more relevant, no differences between cirrhotic and noncirrhotic patients have been so far demonstrated.

Acquired resistance to ICIs is an even more complex phenomenon and is rapidly becoming a hot topic as its occurrence hampers long-term results in patients responding to immunotherapy. Differently from classical chemotherapies and TKIs, ICIs have not a direct an antitumour effect as they act by enhancing the cytotoxicity of the immune system. The acquired resistance to ICIs probably relies on different events, for which, however, dynamic mutations in tumour cells still play a pivotal role. In particular, mutations in genes codifying for target antigens of the HLA system (resulting in a loss of expression of HLA genes on tumour cells) or in genes involved in the interferon signalling may be involved in this phenomenon [74]. While some strategies to overcome these events can be hypothesised (i.e., enhancing the natural killer T-cell response in case of HLA loss), mechanistic and clinical studies are needed to highlight these phenomena further. Interestingly, on ongoing trial for the HCC treatment, combining nivolumab and ABX196 relies on the possibility of activating the natural killer T-cells, potentially overcoming the acquired resistance derived from the HLA loss (NCT03419481).

#### 3.2.5. The Radiological Evaluation of Response

Historically, the RECIST 1.1 [75] have been used as the preferred radiological criteria to assess the response to the systemic drugs for most malignancies, including HCC. However, in the case of HCC, the modified RECIST criteria (mRECIST) assess the response to locoregional treatments and have also been endorsed for the evaluation of systemic therapies [76]. Whether the information provided by the mRECIST in the systemic setting is superior is still a matter of debate [77,78], but the leading regulatory agencies still require a RECIST 1.1-based evaluation.

However, the advent of immunotherapy poses some unique challenges that cannot be addressed by both RECIST1.1 and mRECIST. In early trials of ipilimumab for melanoma, the investigators described an initial disease behaviour meeting the RECIST criteria for progressive disease, followed by marked and durable responses [79]. This pattern was called “pseudoprogression” and was attributed to a delayed response to ICIs and prompted the RECIST working group to propose new immune-related response criteria (iRECIST) [80]. According to these criteria, an increase of the tumour burden or even the appearance of new lesions should be classified as unconfirmed progression (iUPD) [80], and if the patients are clinically stable, ICIs should not be discontinued, and a new imaging assessment should be scheduled in the next 4.-8 weeks. In case of further increase of the tumour burden, radiological progression is confirmed (iCPD), and the treatment should be discontinued. If the tumour remains stable or shrinks, the imaging showing iUPD is regarded as a novel “baseline imaging” for the subsequent evaluations [80].

The combinations of ICIs with either TKIs or anti-VEGF agents could prevent pseudoprogression, and consequently, the applicability of iRECIST for combination therapies is debatable and should be investigated.

## 4. Conclusions

From an expert perspective, accumulating data on ICIs would indicate that these agents can provide answers to some of the current issues in the treatment of HCC.

First, ICIs and their combination provide an objective response rate which is considerably higher than TKIs in monotherapy. These data suggest that pharmacological downstaging strategies for HCC can now be possible. The pertinent implications are manifold. First, patients with intermediate-stage HCC, which are not ideal candidates for transarterial procedures (for instance patients with nodules larger than 6 cm or more than six nodules), due to the low probability of achieving a complete response and for the relatively high risk of hepatic decompensation, could receive upfront systemic treatment, followed by locoregional procedures in case of successful downstaging. Clearly, the feasibility of this strategy and the best cut-offs for tumour size and number have to be defined by future studies. Moreover, the pharmacological downstaging offers can be useful in patients who are a borderline candidate for surgery, and this specific aspect is already being investigated in dedicated trials. Second, a clinically meaningful number of patients treated with ICIs can achieve a durable response, in stark contrast with what occurs with TKIs. At the state-of-art, patients showing an objective response are the most obvious candidates to achieve long-term survival. Thus, identifying combination strategies which augment the biological effects of ICIs will probably be the primary target of future studies. Third, there are no crossed toxicities between TKIs and ICIs; thus, the availability of two different classes of biological agents represents an upmost benefit for patients intolerant to one class.

Nevertheless, there also some open problems which must be necessary to consider, and possibly resolved, to further improve the therapeutic scenario of HCC. Firstly, biomarkers predicting treatment efficacy are still needed. The current lack of such biomarkers designs a scenario affected by a “dilution bias” with a too high percentage of non-responding patients and exposed to the treatment risks. Moreover, this bias adversely affects the cost-effectiveness of ICI treatment. Secondly, despite the increased possibility of achieving long-term responses, disease progression still occurs in most patients, stressing the need to identify agents able to overcome the primary resistance to ICIs and preventing the secondary resistance. Lastly, a word of caution about toxicity: Therapeutic combinations, including ICIs, aimed at increasing the treatment efficacy can also amplify the toxicity. Moreover, since the immune-related AEs are known to occur even after the treatment stop, more long-term data on safety are needed.

## Figures and Tables

**Figure 1 vaccines-08-00578-f001:**
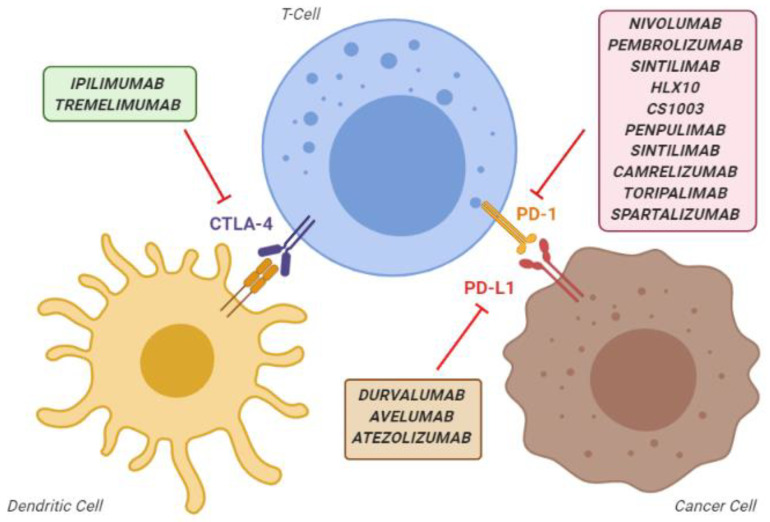
Overview of the main immune checkpoint inhibitors, classified according to their mechanism of action. CTLA-4, cytotoxic T-lymphocyte-associated protein 4; PD-1, programmed death-1 protein; PD-L1, programmed death-ligand 1.

**Table 1 vaccines-08-00578-t001:** Ongoing clinical trials that explore the combination of PD1 inhibitors and intravenous anti-VEGF (vascular endothelial growth factor) agents as treatment for advanced hepatocellular carcinoma.

NCT	Phase	Study Drugs	Line	Primary Endpoint	Estimated End of Trial
NCT03794440	2–3	SINTILIMAB + BEVACIZUMAB BIOSIMILAR	1	OS, ORR	December 2022
NCT03970616	1B/2	DURVALUMAB + TIVOZANIB	1	SAFETY	August 2022
NCT03973112	2	HLX-10+BEVACIZUMAB BIOSIMILAR	1	ORR	June 2022
NCT04393220	2	NIVOLUMAB + BEVACIZUMAB	1	OS, PFS	October 2021
NCT04072679	1	SINTILIMAB + BEVACIZUMAB BIOSIMILAR	1	SAFETY	November 2021

NCT, number of the clinical trial (Clinicaltrials.gov); OS, overall survival; ORR, overall response rate; PFS, progression-free survival.

**Table 2 vaccines-08-00578-t002:** Ongoing clinical trials that explore the combination of PD1 inhibitors and oral tyrosine kinase inhibitors as a treatment for advanced hepatocellular carcinoma.

NCT	Phase	Study Drugs	Line	Primary Endpoint	Estimated End of Trial
NCT04194775	3	CS1003 *+* LENVATINIB vs. LENVATINIB	1	OS, PFS	June 2023
NCT04344158	3	PENPULIMAB *+* ANLOTINIB vs. SORAFENIB	1	OS	December 2024
NCT03755791	3	ATEZOLIZUMAB *+* CABOZANTINIB vs. SORAFENIB	1	OS, PFS	December 2021
NCT03713593	3	PEMBROLIZUMAB *+* LENVATINIB vs. LENVATINIB	1	OS, PFS	May 2022
NCT04411706	2	SINTILIMAB *+* APATINIB *+* CAPECITABINE	1	ORR	June 2022
NCT04042805	2	SINTILIMAB *+* LENVATINIB	1	ORR	August 2024
NCT04444167	2	BISPECIFIC AK104 *+* LENVATINIB	1	ORR	March 2022
NCT04172571	2	PENPULIMAB *+* ANLOTINIB	1	ORR	June 2021
NCT04183088	2	TISLELIZUMAB *+* REGORAFENIB	1	ORR, PFS, SAFETY	March 2025
NCT04052152	2	SINTILIMAB *+* ANLOTINIB	1	ORR, SAFETY	December 2021
NCT03841201	2	NIVOLUMAB *+* LENVATINIB	1	ORR, SAFETY	October 2021
NCT04310709	2	NIVOLUMAB *+* REGORAFENIB	1	ORR	May 2023
NCT04442581	2	PEMBROLIZUMAB *+* CABOZANTINIB	1	ORR	September 2024
NCT04069949	2	TORIPALIMAB *+* SORAFENIB	1	6M-PFS, SAFETY	October 2021
NCT03439891	2	NIVOLUMAB *+* SORAFENIB	1	MTD, ORR	May 2022
NCT04170556	2	NIVOLUMAB *+* REGORAFENIB	2	SAFETY	December 2022
NCT04014101	2	CAMRELIZUMAB *+* APATINIB	2	ORR	October 2021
NCT04170179	2	TORIPALIMAB *+* LENVATINIB *+* CHEMOTHERAPY	1	6M-PFS	December 2020
NCT04401800	1b/2	TISLELIZUMAB *+* LENVATINIB	1	ORR	December 2022
NCT04443309	1b/2	CAMRELIZUMAB *+* LENVATINIB	1	ORR	August 2024
NCT03347292	1	PEMBROLIZUMAB *+* REGORAFENIB	1	DLT, SAFETY	October 2022

NCT, number of the clinical trial (Clinicaltrials.gov); OS, overall survival; PFS, progression-free survival; ORR, overall response rate; MTD, maximum tolerated dose; DLT, dose-limiting toxicities.

**Table 3 vaccines-08-00578-t003:** Ongoing clinical trials of immune checkpoint inhibitors in combination with different drugs.

NCT	Phase	Study Drugs	Line	Primary Endpoint	Estimated End of Trial
NCT03605706	3	CAMRELIZUMAB + FOLFOX4 vs. SORAFENIB OR FOLFOX4	1	OS	December 2021
NCT03519997	2	PEMBROLIZUMAB + BAVITUXIMAB (phosphatidylserine)	1	ORR	April 2022
NCT04050462	2	NIVOLUMAB vs. NIVOLUMAB/BMS-986253 (anti-IL8) vs. NIVOLUMAB/Cabiralizumab (anti-CSF1R)	1	ORR	August 2020
NCT03695250	1–2	NIVOLUMAB + BMS-986205 (IDO1 inhibitor)	1–2	SAFETY, ORR	June 2022
NCT03893695	1–2	NIVOLUMAB + Ascrinvacumab (activin receptor-like kinase 1)	2	DLT	September 2020
NCT03419481	1–2	NIVOLUMAB + ABX196 (invariant Natural Killer T cell agonist)	2	SAFETY	June 2021
NCT03655613	1–2	NIVOLUMAB + APL-101 (cMET inhibitor)	2	DLT	December 2020
NCT02423343	1–2	Nivolumab + galunisertib	2	MTD, SAFETY	July 2020
NCT03241173	1–2	Nivolumab + ipilimumab + INCAGN01949	2	SAFETY, ORR	November 2021
NCT02178722	1–2	Pembrolizumab + epacadostat	2–3	DLT, ORR	August 2020
NCT02795429	1–2	Spartalizumab (+capmatinib)	1	DLT, ORR	October 2020
NCT03095781	1	Pembrolizumab + XL888	2	RP2D	June 2023

NCT, number of the clinical trial (Clinicaltrials.gov); OS, overall survival; ORR, overall response rate; MTD, maximum tolerated dose; DLT, dose-limiting toxicities; RP2D, recommended Phase 2 dose.

**Table 4 vaccines-08-00578-t004:** Ongoing clinical trials exploring checkpoint inhibitors in the adjuvant and neoadjuvant setting.

NCT	Setting	Study Drug(s)	Phase	Primary Endpoint	Estimated End of Trial
NCT03383458	ADJ	NIVOLUMAB vs. PLACEBO	3	RFS	1 June 2025
NCT03867084	ADJ	PEMBROLIZUMAB vs. PLACEBO	3	RFS, OS	1 June 2025
NCT03847428	ADJ	DURVALUMAB *+* BEVACIZUMAB vs. PLACEBO	3	RFS	1 September 2023
NCT04102098	ADJ	ATEZOLIZUMAB *+* BEVACIZUMAB vs. PLACEBO	3	RFS	1 July 2027
NCT03859128	ADJ	TORIPALIMAB vs. PLACEBO	2–3	RFS	1 April 2024
NCT03337841	ADJ	PEMBROLIZUMAB	2	1Y-RFS	1 October 2020
NCT03839550	ADJ	CAMRELIZUMAB *+* APATINIB	2	RFS	1 February 2023
NCT04418401	ADJ	ANTI-PD1 *+* DONAFINIB	2	1Y-RFS	1 June 2023
NCT03630640	ADJ, NADJ	NIVOLUMAB *	2	RFS	1 September 2020
NCT03510871	NADJ	NIVOLUMAB *+* IPILIMUMAB	2	ORR, DOWNSTAGING RATE	1 December 2022
NCT04297202	NADJ	CAMRELIZUMAB *+* APATINIB	2	ORR (10%)	1 December 2021
NCT04297202	NADJ	CAMRELIZUMAB *+* APATINIB	2	ORR (10%)	1 December 2021
NCT04123379	NADJ	NIVOLUMAB *+* CCR2/5-inhibitor vs. NIVOLUMAB *+* ANTI-IL8	2	SAFETY	1 October 2024
NCT03222076	NADJ	NIVOLUMAB	2	SAFETY	1 September 2022
NCT03682276	NADJ	NIVOLUMAB *+* IPILIMUMAB	1–2	DELAY TO SURGERY, SAFETY	1 September 2022
NCT04035876	NADJ	CAMRELIZUMAB *+* APATINIB **	1–2	RFS	1 December 2021
NCT03722875	ADJ	CAMRELIZUMAB	1	RFS	1 March 2020
NCT03383458	ADJ	NIVOLUMAB vs. PLACEBO	1	RFS	1 June 2025
NCT03914352	ADJ	CAMRELIZUMAB ***	n/a	OS, RFS	1 January 2020
NCT04425226	NADJ	PEMBROLIZUMAB *+* LENVATINIB vs. BSC	n/a	RFS, ORR	1 December 2025

NCT, number of the clinical trial (Clinicaltrials.gov); ADJ, adjuvant; RFS, recurrence-free survival; OS, overall survival; 1Y-RFS, 1-year recurrence-free survival; NADJ, neoadjuvant; ORR, overall response rate. * dedicated explicitly to patients undergoing electroporation as ablative technique; ** dedicated explicitly to patients treated with liver transplantation; *** dedicated explicitly to patients with neoplastic portal vein invasion treated with liver resection.

**Table 5 vaccines-08-00578-t005:** Ongoing clinical trials exploring immune checkpoint inhibitors in combination with local therapies.

NCT	Phase	Study Drug(s)	Primary Endpoint	Estimated End of Trial
NCT03778957	3	TACE *+* DURVALUMAB *+* BEVACIZUMAB	PFS	March 2024
NCT04246177	3	TACE *+* PEMBROLIZUMAB *+* LENVATINIB	OS, PFS	December 2029
NCT04229355	3	TACE *+* (SORAFENIB vs. LENVATINIB vs SINTILIMAB)	PFS	December 2022
NCT04268888	2–3	TACE *+* NIVOLUMAB	OS, TTP	June 2026
NCT03753659	2	RFA/MWA/brachytherapy *+* PEMBROLIZUMAB	ORR	September 2023
NCT04297280	2	SINTILIMAB *+* TACE	ORR	May 2023
NCT03857815	2	EBRT *+* SINTILIMAB	PFS	February 2022
NCT03851939	2	HAIC *+* TORIPALIMAB	PFS, ORR	March 2021
NCT03033446	2	SIRT *+* NIVOLUMAB	ORR	December 2019
NCT03482102	2	EBRT *+* DURVALUMAB *+* TREMELIMUMAB	ORR	October 2025
NCT03869034	2	HAIC vs. HAIC *+* PD-1	PFS	March 2022
NCT03572582	2	TACE *+* NIVOLUMAB	ORR	September 2022
NCT03937830	2	TACE *+* DURVALUMAB *+* BEVACIZUMAB	6-M PFS	December 2022
NCT04224636	2	TACE *+* ATEZOLIZUMAB *+* BEVACIZUMAB	2Y-OS	March 2025
NCT02821754	2	ABLATIVE PROCEDURE *+* DURVALUMAB *+* TREMELIMUMAB	PFS	December 2021
NCT04204577	2	TACE *+* CAMRELIZUMAB *+* APATINIB	PFS	November 2023
NCT04220944	2	RFA/TACE *+* SINTILIMAB	PFS	July 2021
NCT04430452	2	EBRT *+* DURVALUMAB *+* TREMELIMUMAB	ORR	August 2024
NCT04191889	2	TAI *+* CAMRELIZUMAB *+* APATINIB	ORR, SAFETY	December 2025
NCT04044313	2	HAIC *+* TORIPALIMAB	PFS	August 2020
NCT04135690	2	HAIC *+* TORIPALIMAB vs. HAIC *+* SORAFENIB	PFS	January 2020
NCT04273100	2	TACE *+* LENVATINIB- *+* PD-1	ORR	June 2021
NCT04150744	2	RFA *+* CARRIZUMAB *+* APATINIB vs CARRIZUMAB *+* APATINIB	PFS	December 2026
NCT04167293	2	EBRT *+* SINTILIMAB	2-Y PFS	October 2025
NCT03316872	2	EBRT *+* PEMBROLIZUMAB	ORR	February 2022
NCT04124991	1–2	SIRT *+* DURVALUMAB	TTP	December 2021
NCT03397654	1B	TACE *+* PEMBROLIZUMAB	SAFETY	December 2020
NCT04104074	1	EBRT *+* SINTILIMAB	SAFETY	December 2020
NCT02837029	1	SIRT *+* NIVOLUMAB	MTD	July 2023

NCT, number of the clinical trial (Clinicaltrials.gov); TACE, transarterial chemoembolisation; PFS, progression-free survival; OS, overall survival; TTP, time to progression; RFA, radiofrequency ablation; MWA, microwave ablation; ORR, overall response rate; EBRT, external beam radiation therapy; HAIC, hepatic artery infusion chemotherapy; SIRT, selective intraarterial radiation treatment; MTD, maximum tolerated dose.

**Table 6 vaccines-08-00578-t006:** Comparison of the main efficacy and safety data from clinical trials of immune checkpoint inhibitors in monotherapy or in combination with other agents.

Parameter/Endpoint	Nivolumab	Pembrolizumab	Atezolizumab + Bevacizumab	Pembrolizumab + Lenvatinib	Nivolumab + Ipilimumab
Median OS (months)	16.4	13.9	>17.0	22.0	12.2–22.5
Median PFS (months)	3.7	3.0	6.8	8.6	not reported
ORR	15%	18.3%	27.3%	36.6%	31–32%
G ≥3 AEs	22%	46.3%	56.5%	67% *	37%
Discontinuation rate for AEs	4%	17.2%	15.5%	not reported	2–18%

OS, overall survival; PFS, progression-free survival; ORR, objective response rate; AEs, adverse events. * Including five Grade 5 events.

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
