# Peer review of "Immunotherapy with Checkpoint Inhibitors for Hepatocellular Carcinoma: Where Are We Now?"

_vaccines, 2020, doi:10.3390/vaccines8040578_

Round 1

Reviewer 1 Report

Major comments:

  • Much of the text and trial results could be included in the existing tables and would convey the results more succinctly and read more fluidly.
  • The review presents a re-statement of facts from clinical trials and clinically relevant considerations for ICI in HCC. While this information is a helpful summary, it doesn't add much more than is already readily searchable. Perhaps the review may be more impactful if written from the perspective of an expert in the field that has insight into the challenges in treatment, why ICI is relevant in HCC treatment, the successes and failures of the past, ongoing debates amongst experts, leading hypotheses about clinical findings, the learnings of these trials, and future directions to improve treatment. While these topics are covered to some extent in the review, more in depth discussion and a restructuring of the article may be helpful to improve the fluidity of the narrative and readability. 

Minor comments:

  • Definition of Checkpoint inhibitors in the introduction reads as if PD1, PDL-1, CTLA-4 make up the entire checkpoint field and are the only ones. Consider rephrasing and defining how checkpoint inhibitors broadly act.
  • Some anecdotal comments are made throughout the text (ie. "....closed their enrolment earlier than initially scheduled time, suggesting a high accrual rate.) Text should stick to known / factual statements. 
  • While the authors discuss a variety of cellular etiologies for lack of ICI efficacy, commentary about the significance of any one of these mechanisms in HCC specifically could be included. Many of these cellular mechanisms for ICI failure discussed are true across several tumour models. Discussion of the mechanisms or other existing hypotheses in the context of HCC biology specifically may provide greater perspective about the disease's immunosuppressive microenvironment or other hurdles that need to be overcome to improve future therapies.

Author Response

REVIEWER 1

Major comments:

  • Much of the text and trial results could be included in the existing tables and would convey the results more succinctly and read more fluidly.

The existing Tables actually contain information about the ongoing - but still not concluded – trials. Therefore, it is not possible to include the results of the concluded trials in them. Also, we feel that creating new tables would result in a very high number of total tables, making the text very difficult to explore.

  • The review presents a re-statement of facts from clinical trials and clinically relevant considerations for ICI in HCC. While this information is a helpful summary, it doesn't add much more than is already readily searchable. Perhaps the review may be more impactful if written from the perspective of an expert in the field that has insight into the challenges in treatment, why ICI is relevant in HCC treatment, the successes and failures of the past, ongoing debates amongst experts, leading hypotheses about clinical findings, the learnings of these trials, and future directions to improve treatment. While these topics are covered to some extent in the review, more in depth discussion and a restructuring of the article may be helpful to improve the fluidity of the narrative and readability. 

We thank the reviewer for these suggestions that offer us the opportunity to greatly improve the message of the article. We added our experts’ perspective both in the Introduction and in the Conclusions.

Introduction: This priority is determined by multiple unmet needs. First, the disease control rate obtained with TKIs rarely exceeds 50-60%, with an objective response rate constantly <10%. Indeed, a relatively high number of patients show a progression at the first imaging evaluation with the practical failure of obtaining tumour downstage or neoadjuvant/adjuvant strategies. Indeed, a high number of patients progress after the first imaging evaluation with the practical failure of obtaining tumour downstage or neoadjuvant/adjuvant strategies. Second, even patients achieving a disease control will experience a relatively short progression-free survival, limiting the possibility of achieving long-term survivals in a sizeable number of patients. Third, most patients will experience TKI-related adverse events (AEs) impairing their quality of life and the adherence to the treatment strategy. Moreoover, as the safety profile of TKIs widely overlaps between different molecules, patients who are intolerant to sorafenib are theoretically prone to experience similar AEs if treated with other TKIs. As an example, the registrative trial of regorafenib explicitly excluded patients who did not tolerate sorafenib at the reduced dose of 400 mg/day. Consequently, most regulatory agencies approved the use of regorafenib only for patients who progressed under sorafenib, but not for those intolerant to the drug. 

Conclusions: From an expert perspective, accumulating data on ICIs would indicate that these agents can provide answers to some of the current issues in the treatment of HCC.

First, ICIs and their combination provide an objective response rate which is considerably higher than TKIs in monotherapy. These data suggest that pharmacological downstaging strategies for HCC can now be possible. The pertinent implications are manifold. First, patients with intermediate stage HCC which are not ideal candidates for transarterial procedures (for instance patients with nodules larger than 6 cm or more than 6 nodules) due the low probability of achieving a complete response and for the relatively high risk of hepatic decompensation, could receive upfront systemic treatment, followed by locoregional procedures in case of successful downstaging. Clearly, the feasibility of this strategy and the best cut offs for tumour size and number have to be defined by future studies. Moreover, the pharmacological downstaging offers can be useful in patients who are borderline candidate for surgery, and this specific aspect is already being investigated in dedicated trials. Second, a clinically meaningful number of patients treated with ICIs can achieve a durable response, in a stark contrast with what occurs with TKIs. At the state-of-art, patients showing an objective response are the most obvious candidates to achieve a long survival. Thus, identifying combination strategies which augment the biological effects of ICIs will probably be the primary target of future studies. Third, there are no crossed toxicities between TKIs and ICIs, thus the availability of two different classes of biological agents represents an upmost benefit for patients intolerant to one class.

Nevertheless, there also some open problems which must be necessary to consider, and possibly resolved, to further improve the therapeutic scenario of HCC. Firstly, biomarkers predicting treatment efficacy are still needed. The current lack of such biomarkers designs a scenario affected by a “dilution bias” with a too high percentage of non-responding patients and exposed to the treatment risks. Moreover, this bias adversely affects the cost-effectiveness of ICI treatment. Secondly, despite the increased possibility of achieving long-term responses, disease progression still occurs in most patients, stressing the need to identify agents able to overcome the primary resistance to ICIs and preventing the secondary resistance. Lastly, a word of acution about toxicity. Indeed, therapeutic combinations including ICIs aimed at increasing the treatment efficacy can also amplify the toxicity. Moreover, since the immune-related AEs are known to occur even after the treatment stop, more long term data on safey are needed.

Minor comments:

  • Definition of Checkpoint inhibitors in the introduction reads as if PD1, PDL-1, CTLA-4 make up the entire checkpoint field and are the only ones. Consider rephrasing and defining how checkpoint inhibitors broadly act.

We agree that our sentence was misleading. The sentence has been rephrased as follows:

Introduction: Immune checkpoints are a normal part of the immune system. Their role is to prevent hyperimmune responses leading to tissue damage. The most known immune checkpoint are PD-1 and Cytotoxic T-lymphocyte-associated protein 4 (CTLA-4).

  • Some anecdotal comments are made throughout the text (ie. "....closed their enrolment earlier than initially scheduled time, suggesting a high accrual rate.) Text should stick to known / factual statements. 

The pertinent sentences have been corrected.

  • While the authors discuss a variety of cellular etiologies for lack of ICI efficacy, commentary about the significance of any one of these mechanisms in HCC specifically could be included. Many of these cellular mechanisms for ICI failure discussed are true across several tumour models. Discussion of the mechanisms or other existing hypotheses in the context of HCC biology specifically may provide greater perspective about the disease's immunosuppressive microenvironment or other hurdles that need to be overcome to improve future therapies

We appreciate the reviewer’s suggestion. We added the following consideration to the text:

Currently, there are no data suggesting that the cirrhotic microenviroment actually affects the efficacy of ICIs. Indeed, when the first trials of immunotherapy for HCC were designed, this hypothesis (deriving from pre-clinical experiences) was considered, but subsequent clinical data showed a similar efficacy both in viral and nonviral patients. Even more relevant, no differences between cirrhotic and noncirrhotic patients have been so far demonstrated.

Reviewer 2 Report

Tovoli et al. present an impressive up-to-date review on the current standing of ICI- and ICI combination therapies for HCC. The paper is of interest for a wide audience, including clinicians and cancer biologists. The clear writing style and the well-placed tables also allow patients to access some of the information. Please find my minor comments below.

Results:

  1. line 116-132 (Keynote 240 trial): Is there an explanation for the much higher AE and discontinuation rate with Pembrolizumab? Is it related to the lower OS of patients compared to the Checkmate 459 trial (lines 97-115)?
  2. line 132-133: What does the discontinuation of the placebo treatment arm mean? Did these patients receive chemotherapy?
  3. The Author might consider inserting a table to compare AE and withdrawal rate of the different ICI agents and combination therapies.
  4. To be more patient-centered, the Authors/Editors may consider adding a short lay summary.
  5. Text references 61 and after are in italics.

Author Response

REVIEWER 2

Tovoli et al. present an impressive up-to-date review on the current standing of ICI- and ICI combination therapies for HCC. The paper is of interest for a wide audience, including clinicians and cancer biologists. The clear writing style and the well-placed tables also allow patients to access some of the information.

Thank you for these  favorable comments

  • line 116-132 (Keynote 240 trial): Is there an explanation for the much higher AE and discontinuation rate with Pembrolizumab? Is it related to the lower OS of patients compared to the Checkmate 459 trial (lines 97-115)?

The answer is not easy, as no definite conclusions can be reached reading the published data. Probably, the different clinical setting is the main reasons. In fact, the Keynote 240 trial enrolled patients in the second-line setting, which usually include patients which are more frail than those of the frontline setting. We added to the text the following considerations:

“Interestingly, the AEs and discontinuation rates were more frequent with pembrolizumab than with nivolumab, probably underlining the frailer characteristics of patients enrolled in the second-line setting. Pertinently, in the Checkmate 459 trial a high proportion of placebo patients discontinued the treatment due to the occurrence of AEs which, of course, were caused by the underlying clinical condition. ”

  • line 132-133: What does the discontinuation of the placebo treatment arm mean? Did these patients receive chemotherapy?

The Keynote-240 trial had a double-blind design. So, patients in the placebo arm discontinued the treatment in the case that the investigator (who was blinded to the content of the vials) thought that the AE was due to the treatment. The relatively high proportion of patients who discontinued in the placebo arm supports our previous hypothesis of relatively frail patients, since AEs leading to treatment discontinuation were not actually drug-related but probably due to their underlying condition (see previous point). We mentioned in the manuscript: “the relatively low number of patients enrolled, and the high number of placebo patients (47.4%) receiving post-study chemotherapies were blamed for the failure of the trial.”

  • The Author might consider inserting a table to compare AE and withdrawal rate of the different ICI agents and combination therapies.

Thanks for this valuable suggestion. A dedicated table (Table 6) has been added.

  • To be more patient-centered, the Authors/Editors may consider adding a short lay summary.

We agree with this suggestion, but Vaccines template does not contemplate a Lay Summary. We will add this part should the Editors allow such a possibility.

  • Text references 61 and after are in italics.

This error has been fixed.

Reviewer 3 Report

In this review, Tovoli and colleagues describe progress for immune checkpoint inhibitors (ICIs) for hepatocellular carcinoma (HCC). Overall  this review is systematic and comprehensive, though it would benefit from a more tempered tone in places and contains several small, but important errors that must be corrected prior to publication. 

Specific comments are detailed below: 

-The authors begin their abstract with the phrase "Immune checkpoint inhibitors (ICIs) are destined to play a central role in the systemic 11 treatment of hepatocellular carcinoma (HCC)". The evidence supporting ICIs is weak at best, and this is extremely speculative and baseless. This should be amended to reflect the current state of ICIs in HCC. A suggestion is that "Immune checkpoint inhibitors are beginning to show early promise in the clinical management of hepatocellular carcinoma", etc. 

-In the introduction authors repeatedly use superfluous, non-professional language to describe the results of clinical trials. For instance, they describe the results of a trial comparing nivolumab to sorafinib as a "crushing disappointment". Another drug is described as "substantially challenging" sorafinib. They also use terms as "outclassed" and "brilliant success" to describe a trial that at best was a modest success, as the combination of atezolizumab and bevacizumab had a 12 month survival or 67.2% compared to 54.6% with sorafinib, with a median PFS of 6.8 v 4.3 months. Thus, their language here is extremely misleading.

-These issues are evident throughout the manuscript, with other examples including "the great expectations deriving from these results were subsequently frustrated by the failure of two large Phase 3 trials", "Efficacy data showed a whopping OS of 22.8 months in Group A, "interpretation of skin toxicities can be tricky", or "The results reported by Sangro et al. [14] in their pivotal trial with tremelimumab were accompanied by mixed feeling". 

. The entire manuscript should be carefully edited to more objectively report the results of all clinical data and avoid these non-academic descriptors. 

-The "Materials and Methods" section for a review article is needless and of little value. 

-The "Results" subheading in a review article is very odd. I would suggest eliminating the above "Materials and Methods" section and "Results" subheading and take on a more conventional structure for a review article. 

-Enrollment is misspelled on line 153

-Regarding the section beginning on line 494, the role for TGFB signaling in resistance to ICIs has little to do with EMT. This section ignores the emerging and well-established roles of TGFB as a critical immunomodulator. For instance, TGFB is a potent inducer of  suppressive regulatory T-cells, and has potent effects on CD8+ T-cells, causing them to remain refractory from full activation. The authors should expand on this topic including several references discussing the many roles of TGFB in the tumor microenvironment, as what is stated is a gross oversimplification and incredibly misleading. 

Author Response

REVIEWER 3

-The authors begin their abstract with the phrase "Immune checkpoint inhibitors (ICIs) are destined to play a central role in the systemic 11 treatment of hepatocellular carcinoma (HCC)". The evidence supporting ICIs is weak at best, and this is extremely speculative and baseless. This should be amended to reflect the current state of ICIs in HCC. A suggestion is that "Immune checkpoint inhibitors are beginning to show early promise in the clinical management of hepatocellular carcinoma", etc.

This sentence has been modified as suggested.

-In the introduction authors repeatedly use superfluous, non-professional language to describe the results of clinical trials. For instance, they describe the results of a trial comparing nivolumab to sorafinib as a "crushing disappointment". Another drug is described as "substantially challenging" sorafinib. They also use terms as "outclassed" and "brilliant success" to describe a trial that at best was a modest success, as the combination of atezolizumab and bevacizumab had a 12 month survival or 67.2% compared to 54.6% with sorafinib, with a median PFS of 6.8 v 4.3 months. Thus, their language here is extremely misleading.

Thanks for your suggestion. This sentence has been modified as suggested.

-These issues are evident throughout the manuscript, with other examples including "the great expectations deriving from these results were subsequently frustrated by the failure of two large Phase 3 trials", "Efficacy data showed a whopping OS of 22.8 months in Group A, "interpretation of skin toxicities can be tricky", or "The results reported by Sangro et al. [14] in their pivotal trial with tremelimumab were accompanied by mixed feeling".

Thanks for pointing out these problems. These sentences have been modified as suggested.

. The entire manuscript should be carefully edited to more objectively report the results of all clinical data and avoid these non-academic descriptors.

We performed all the modifications suggested by the reviewer.

-The "Materials and Methods" section for a review article is needless and of little value.

We understand the reviewer’s position. However, the journal instructs to maintain this format. From the Instruction for Authors:  “Structured reviews and meta-analyses should use the same structure as research articles and ensure they conform to the PRISMA guidelines”. In any case, we feel that the searching strategy will help the readers in verifying our sources of information.

-The "Results" subheading in a review article is very odd. I would suggest eliminating the above "Materials and Methods" section and "Results" subheading and take on a more conventional structure for a review article.

We understand the reviewer’s position. However, the journal instructs to maintain this format. From the Instruction for Authors:  “Structured reviews and meta-analyses should use the same structure as research articles and ensure they conform to the PRISMA guidelines”. In any case, we feel that the searching strategy and its results will help the readers in verifying our sources of information.

-Enrollment is misspelled on line 153

Thanks for pointing out this error.

-Regarding the section beginning on line 494, the role for TGFB signaling in resistance to ICIs has little to do with EMT. This section ignores the emerging and well-established roles of TGFB as a critical immunomodulator. For instance, TGFB is a potent inducer of  suppressive regulatory T-cells, and has potent effects on CD8+ T-cells, causing them to remain refractory from full activation. The authors should expand on this topic including several references discussing the many roles of TGFB in the tumor microenvironment, as what is stated is a gross oversimplification and incredibly misleading.

We fully agree with the reviewer about this point. TGF-beta roles are manifold and our sentence could provide misleading information. We updated this part as follows:

The role of tumour growth factor beta (TGF beta) is also of particular interest. This multifunctional cytokine plays multiple key activity, because of its role in immune and stem cell regulation and differentiation [67,68].  In many cancer cells,  the TGF-β signaling is disrupted [69]. As a consequence, TGF-β is no longer able to downregulated the cell cycle, causing a simultaneous proliferation of both cancer and surrounding stromal cells in the setting of an immunosuppressive and pro-angiogenic microenviroment [70]. Additionally, TGF-β can convert effector T-cells into regulatory T-cells [71]and exert inhibitory effects on B-cells [72] , turning off the inflammatory reaction and favouring tumour immune escape. Investigation on the combination of nivolumab and the TGF-beta inhibitors galunisertib and ascrinvacumab (NCT02423343, NCT03893695) is currently in progress and will provide valuable information about the ability of these combinations in overcoming resistance to ICIs

Round 2

Reviewer 1 Report

Authors were able to make significant improvements to the manuscript, addressed initial comments that were suggested, and clarified ambiguous pharases in the article.